# Plasma metabolomic profiles associated with mortality and longevity in a prospective analysis of 13,512 individuals

Fenglei Wang [1], Anne-Julie Tessier [1], Liming Liang [2,3], Clemens Wittenbecher[1,4], Danielle E. Haslam[1,5], Gonzalo Fernández-Duval [6], A. Heather Eliassen [1,2,5], Kathryn M. Rexrode [7,8], Deirdre K. Tobias[1,7], Jun Li [1,7], Oana Zeleznik [5], Francine Grodstein[9], Miguel A. Martínez-González[1,6,10], Jordi Salas-Salvadó [10,11,12], Clary Clish [13], Kyu Ha Lee[1,2,3], Qi Sun [1,2,5], Meir J. Stampfer[1,2,5], Frank B. Hu[1,2,5,16] ✉ & Marta Guasch-Ferré [1,14,15,16] ✉

Experimental studies reported biochemical actions underpinning aging processes and mortality, but the relevant metabolic alterations in humans are not well understood. Here we examine the associations of 243 plasma metabolites with mortality and longevity (attaining age 85 years) in 11,634 US (median follow-up of 22.6 years, with 4288 deaths) and 1878 Spanish participants (median follow-up of 14.5 years, with 525 deaths). We find that, higher levels of N2,N2-dimethylguanosine, pseudouridine, N4-acetylcytidine, 4-acetamidobutanoic acid, N1-acetylspermidine, and lipids with fewer double bonds are associated with increased risk of all-cause mortality and reduced odds of longevity; whereas L-serine and lipids with more double bonds are associated with lower mortality risk and a higher likelihood of longevity. We further develop a multi-metabolite profile score that is associated with higher mortality risk. Our findings suggest that differences in levels of nucleosides, amino acids, and several lipid subclasses can predict mortality. The underlying mechanisms remain to be determined.

Global average life expectancy increased by 5.5 years between 2000 and 2016, the fastest increase since the 1960s[1]. This increase could be the result of a number of factors such as improvements in living standards and medical treatment and a substantial reduction in smoking[2], among others[3]. These factors have contributed not only to the increased overall life expectancy, but also to the fact that living beyond 80 years is becoming increasingly common[1]. However, alterations in metabolism underlying mortality and aging process are not completely understood.

Metabolomics offers a novel avenue for the discovery of novel biomarkers linked to aging and disease. Emerging evidence has shown promise in identifying metabolites associated with mortality and longevity in animal models and humans[4]. Various metabolites including acylcarnitines, amino acids, phospholipids, and purine/pyrimidines have been associated with all-cause mortality or longevity[5–10]. In addition, previous studies have also identified metabolites, including acylcarnitines and sphingomyelins, that are associated with age[11]. These metabolites may shed light on pathways to survival and longevity in humans. Data from genetic and experimental studies support causal roles for some of these metabolites in biological aging[12]. However, data in human populations are constrained by small sample sizes, short follow-up, and a limited number of metabolites profiled. Identifying robust metabolite predictors of both long-term mortality risk and longevity could help to identify new preventive strategies.

In this work, we perform a high-throughput metabolomics-based investigation in three well-phenotyped cohorts of middle-aged to older adults: the Nurses' Health Study (NHS), NHSII, and the Health Professionals Follow-up Study (HPFS). We first identify circulating metabolites that are associated with all-cause and cause-specific mortality risk and longevity (living to age 85 years or older). Subsequently, we construct a multi-metabolite profile score and examine its associations with mortality and longevity. Finally, we independently replicate these associations in the PREvención con DIeta MEDiterránea (PREDIMED) study.

## Results

### Characteristics of the study participants

Our primary analyses included 11,634 participants who had been selected for 13 prior nested case-control sub-studies on metabolomics from the NHS, NHSII, and HPFS (Fig. 1 and Supplementary Table 1). Participants were predominantly white and female, middle-aged (mean age $54.3 \pm 9.0$ years), and had an average BMI of $25.7 \pm 4.9$ kg/m$^2$ (Table 1). The replication analyses included 1878 participants from the PREDIMED study. They were older (mean age $67.1 \pm 6.1$ years) and had a higher BMI ($29.9 \pm 3.6$ kg/m$^2$) and a higher prevalence of diabetes, hypertension, and hypercholesterolemia than participants from NHS/NHSII/HPFS (Table 1). Compared to participants who survived during follow-up, participants who died were generally older at blood draw and less likely to be female, more likely to smoke, drink more alcohol, and have diagnosed metabolic diseases in both NHS/NHSII/HPFS and PREDIMED (Supplementary Table 2). In contrast,

participants who achieved longevity were more likely to be female, less likely to smoke, and had a lower alcohol intake compared to those who did not (Supplementary Table 3).

### Metabolome-wide associations for all-cause, cardiovascular, and cancer mortality

We first conducted a metabolome-wide association analysis in NHS/NHSII/HPFS. During a median follow-up of 22.6 years after blood collection, we documented 4288 deaths, including 867 cardiovascular and 1074 cancer deaths. After adjusting for potential confounders, 75 metabolites were positively associated and 32 inversely associated with all-cause mortality at a false discovery rate (FDR) < 5% (Fig. 2a and Supplementary Data 1). Metabolites that were most positively associated with mortality were C16:1 cholesterol esters (CE), N2,N2-dimethylguanosine, ureidopropionic acid, C16:0 Ceramide (d18:1), pseudouridine, 4-acetamidobutanoic acid, and phosphatidylcholines (PCs), phosphatidylethanolamines (PEs), diacylglycerols (DAGs), and triacylglycerols (TAGs) with low saturation. Metabolites most negatively associated with mortality were L-serine, C22:6 CE, piperine, bilirubin, L-threonine, 4-hydroxy-3-methylacetophenone, and several highly unsaturated plasmalogens, PCs, and TAGs. Given that participants were selected from case-control sub-studies, we further conducted an analysis stratified by case/control groups. The associations for these 107 metabolites were generally similar among cases and controls (Fig. 2b). We also found similar associations across different baseline age (<55 or ≥55 years) and sex (Supplementary Fig. 1a and 1b). Results remained consistent when limiting follow-up to the first 10 or

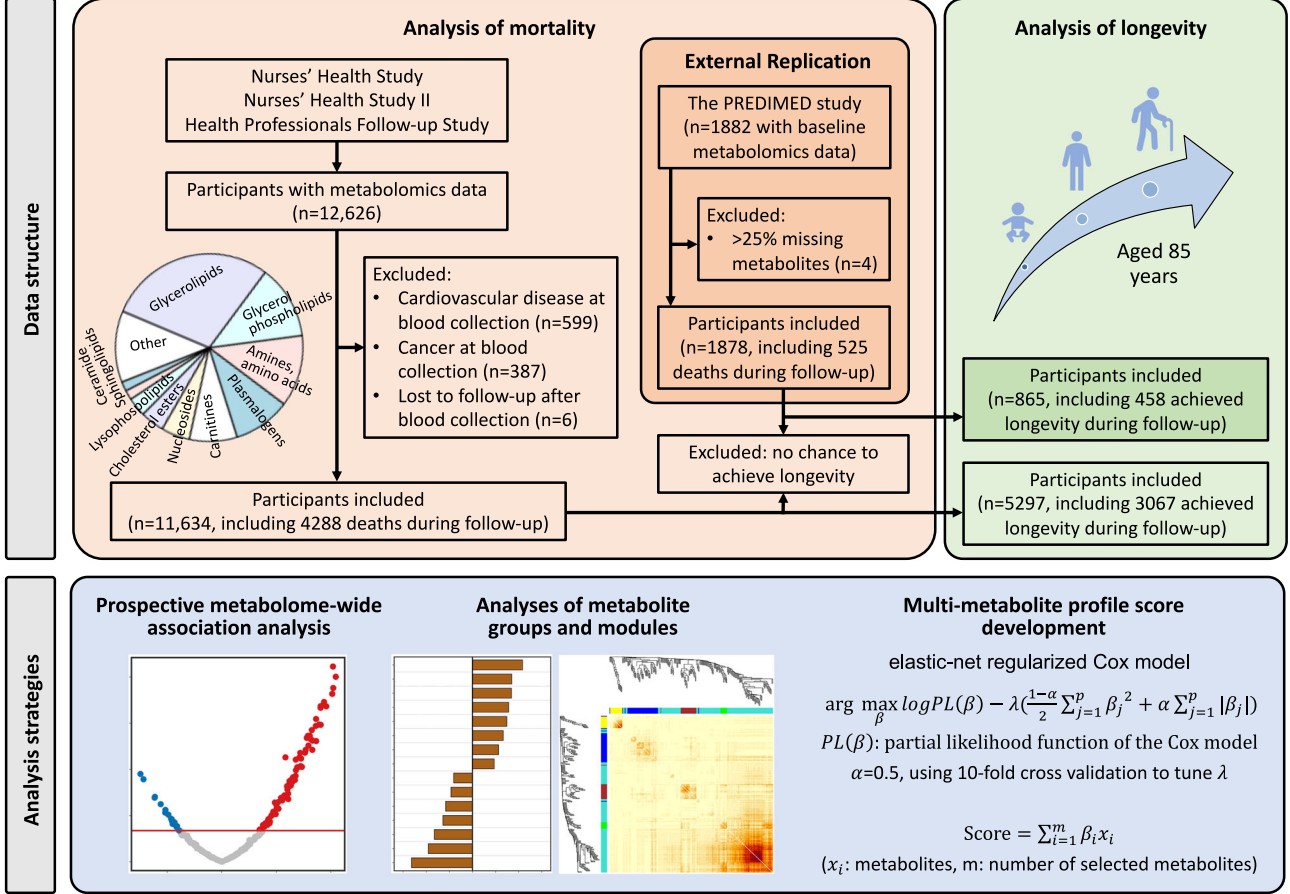

**Fig. 1 | Schematic of the study design.** We examined the associations of circulating metabolites with all-cause, cardiovascular, and cancer mortality among participants with available metabolomics data in Nurses' Health Study, Nurses' Health Study II, and Health Professionals Follow-up Study. We further assessed the associations between metabolites and longevity, defined as reaching age 85 years. In addition, we developed a multi-metabolite profile score for all-cause mortality and examined its association with mortality and longevity. Results were externally replicated in PREDIMED.

## Table 1 | Baseline characteristics of participants

| | NHS/NHSII/HPFS | | | | PREDIMED (n = 1878) |
|---|---|---|---|---|---|
| | NHS (n = 6883) | NHSII (n = 3131) | HPFS (n = 1620) | Overall (n = 11,634) | |
| Age, years | 56.7 (6.8) | 44.6 (4.5) | 62.5 (8.2) | 54.3 (9.0) | 67.1 (6.1) |
| Female/male, %/% | 100/0 | 100/0 | 0/100 | 86.1/13.9 | 57.5/42.5 |
| Ethnicity (white), % | 96.2 | 97.3 | 94.7 | 96.3 | 100 |
| Fasting at blood collection, % | 75.5 | 72.1 | 59.1 | 72.3 | 100 |
| Body mass index, kg/m$^2$ | 25.7 (4.7) | 25.8 (5.9) | 25.6 (3.1) | 25.7 (4.9) | 29.9 (3.6) |
| Physical activity (MET-hours/week) | 15.9 (22.4) | 17.8 (21.8) | 31.9 (29.5) | 18.7 (24.0) | 4.1 (4.0) |
| Smoking status, % | | | | | |
| Never | 46.8 | 66.9 | 47.0 | 52.2 | 59.3 |
| Past | 40.2 | 24.8 | 48.8 | 37.3 | 24.8 |
| Current | 13.0 | 8.3 | 4.2 | 10.5 | 15.9 |
| Multivitamin use, % | 64.2 | 76.6 | 58.6 | 66.8 | / |
| Medical history, % | | | | | |
| Diabetes | 3.1 | 1.6 | 3.3 | 2.8 | 26.7 |
| Hypertension | 26.6 | 11.1 | 26.7 | 22.5 | 87.3 |
| Hypercholesterolemia | 27.9 | 24.9 | 34.9 | 28.0 | 76.8 |
| Total energy intake, kcal/day | 1781 (516) | 1831 (546) | 2010 (605) | 1827 (543) | 2317 (602) |
| Alcohol intake, g/day | 5.8 (10.2) | 3.5 (6.3) | 11.0 (14.8) | 5.9 (10.4) | 9.6 (15.6) |
| Alternate Healthy Eating Index (not including alcohol) | 47.3 (9.9) | 45.6 (10.0) | 48.2 (11.1) | 47.0 (10.2) | / |

Values are means (SDs) for continuous variables and percentages for categorical variables.

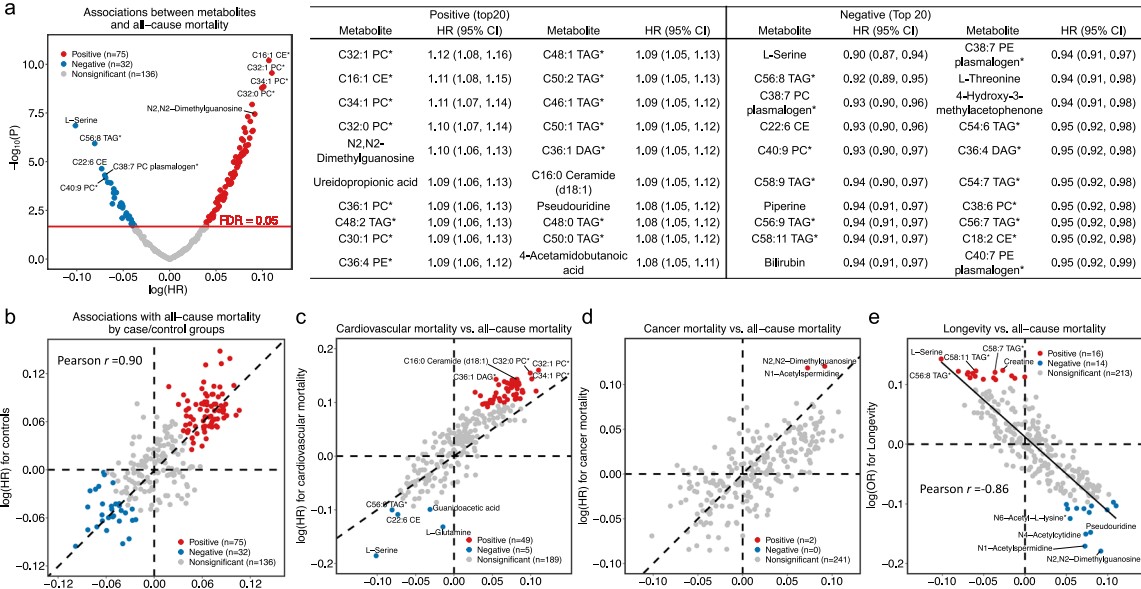

**Fig. 2 | Metabolome-wide associations for all-cause mortality, cardiovascular mortality, cancer mortality, and longevity. a** Volcano plot for the associations between metabolites and all-cause mortality and HR (95% CI) per 1-SD increment for top 40 metabolites (20 positive and 20 negative). **b** Scatter plot for analysis stratified by case/control groups. **c** Scatter plot for log(HR) (β coefficient from Cox proportional hazard regression) for cardiovascular mortality versus log(HR) for all-cause mortality. **d** Scatter plot for log(HR) for cancer mortality versus log(HR) for all-cause mortality. **e** Scatter plot for log(OR) (β coefficient from logistic regression) for longevity versus log(HR) for all-cause mortality. All results were from the multivariable Cox model stratified by study cohorts, original sub-studies, and the case/control status in the original sub-study and adjusted for age, fasting status, body mass index, race, multivitamin use, smoking status, physical activity, diabetes, hypertension, antihypertensive medication use, hypercholesterolemia, lipid-lowering medication use, total energy intake, alcohol intake, and Alternate Healthy Eating Index. Metabolites with * indicate representative names. Source data are provided as a Source Data file.

20 years, when excluding participants diagnosed with cardiovascular disease (CVD) or cancer within the first 4 years after blood draw, and when conducting complete case analysis without imputation for missing metabolite data (Supplementary Fig. 1c–f).

A total of 54 metabolites were associated with cardiovascular mortality after adjustment for multiple testing (FDR < 0.05) (Fig. 2c).

Most of them were also identified in the analysis for all-cause mortality, including C16:0 Ceramide (d18:1), 4-acetamidobutanoic acid, and several PCs, which were positively associated with mortality, and L serine, C22:6 CE, and highly unsaturated TAGs, which were negatively associated with mortality (Supplementary Data 1). Additionally, we found that L-glutamine and guanidoacetic acid were associated with

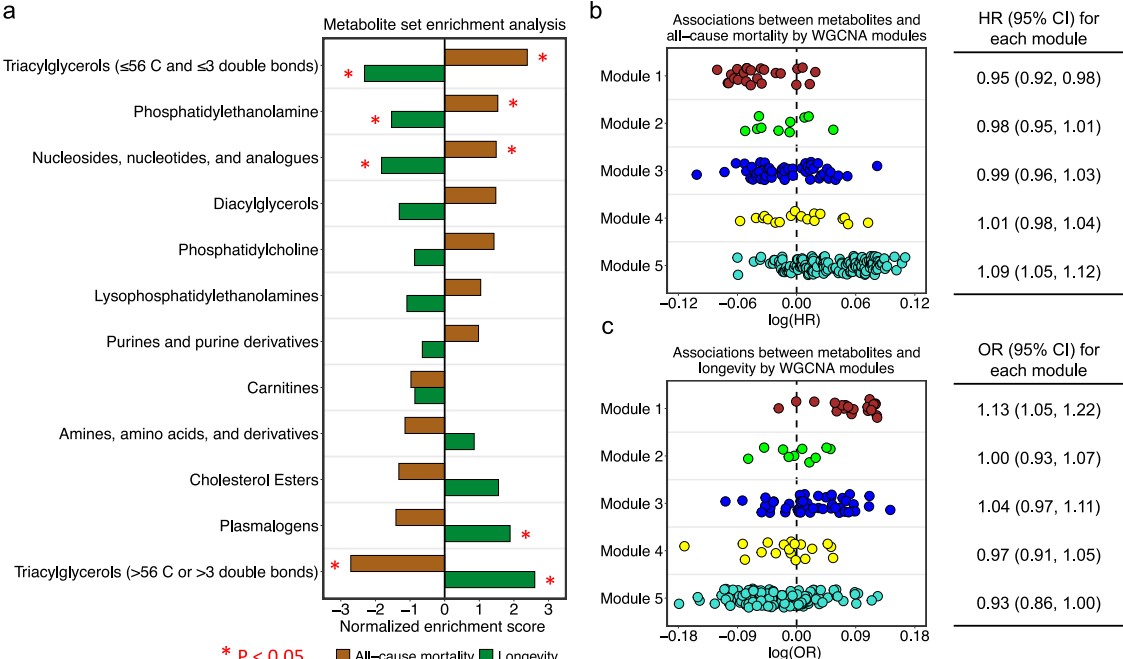

**Fig. 3 | Associations of metabolite groups and modules with all-cause mortality and longevity. a** Association between knowledge-based metabolite groups and all-cause mortality. Metabolite Set Enrichment Analysis was used to estimate enrichment scores based on estimates from multivariable Cox models. **b** Metabolome-wide association for all-cause mortality by WGCNA derived modules, and association for each module, estimated using multivariable Cox models. **c** Metabolome-wide association for longevity by WGCNA derived modules, and association for each module, estimated using multivariable logistic regression models. All results were from the multivariable model stratified by study cohorts, original sub-studies, and the case/control status in the original sub-study, and adjusted for age, fasting status, body mass index, race, multivitamin use, smoking status, physical activity, diabetes, hypertension, antihypertensive medication use, hypercholesterolemia, lipid-lowering medication use, total energy intake, alcohol intake, and Alternate Healthy Eating Index. Abbreviation: WGCNA, weighted gene co-expression network analysis. Source data are provided as a Source Data file.

lower risk of cardiovascular mortality. Only two metabolites, N2,N2-dimethylguanosine and N1-acetylspermidine were associated with cancer mortality at FDR < 0.05 (Fig. 2d), and neither was associated with cardiovascular mortality.

To examine whether incident CVD or cancer is mediating the association between metabolites and mortality, we performed a mediation analysis. We observed that associations for several metabolites with cardiovascular mortality could be partly attributed to CVD diagnosed during follow-up (Supplementary Fig. 2 and Supplementary Data 2). The largest attributable proportion was 59.1% for 4-acetamidobutanoic acid, whose association with higher cardiovascular mortality risk was attenuated the most after adjusting for incident CVD. Associations for C16:0 Ceramide (d18:1), L-serine, and PEs and glycerolipids with low saturation were also attenuated after incident CVD adjustment, with attributable proportion ranging from 25.4 to 52.4%. Associations with all-cause and cancer mortality were not or only slightly mediated by incident CVD or cancer (Supplementary Data 2 and 3).

**Metabolite groups and modules associated with mortality**

To identify metabolite groups/modules associated with mortality, we conducted metabolite set enrichment analysis (MSEA) and weighted gene co-expression network analysis (WGCNA). Four knowledge-based metabolite groups were associated with all-cause mortality in the MSEA analysis (Fig. 3a and Supplementary Data 4). Triacylglycerols with ≤56 carbon and ≤3 double bonds; phosphatidylethanolamines; and nucleosides, nucleotides, and analogues were associated with increased risk of all-cause mortality, while triacylglycerols with >56 carbon or >3 double bonds were associated with lower all-cause mortality. These associations were also observed for cardiovascular mortality and cancer mortality

(Supplementary Fig. 3a). WGCNA identified five metabolite modules (Supplementary Fig. 4 and Supplementary Data 4), two of which were associated with all-cause mortality (Fig. 3b). Module 1, characterized by highly unsaturated TAGs, PCs, and plasmalogens, was associated with lower all-cause mortality with an HR of 0.95 (95% CI: 0.92, 0.98) per 1-SD increment in the module score. Module 5, characterized by nucleosides and TAGs with ≤56 carbon and ≤3 double bonds, was associated with higher all-cause mortality (HR = 1.09; 95% CI: 1.05, 1.12). Module 5 was also associated with an increased risk of cardiovascular mortality (Supplementary Fig. 3b). No modules were associated with cancer mortality (Supplementary Fig. 3c).

**Multi-metabolite profile score for mortality**

To summarize the metabolomic profile associated with all-cause mortality into a score, we performed an elastic net regression, which identified a panel of 75 metabolites to construct the score (Supplementary Data 5). Metabolites in this panel included those observed from the metabolome-wide association analysis, such as C16:1 CE, C16:0 Ceramide (d18:1), ureidopropionic acid, 4-acetamidobutanoic acid, and N2,N2-dimethylguanosine for positive associations and L-Serine, L-threonine, bilirubin, piperine, and 4-hydroxy-3-methylacetophenone for negative associations. A higher multi-metabolite profile score based on these 75 metabolites was associated with increased risk of all-cause (HR per 1-SD increment = 1.26; 95% CI: 1.22, 1.31), cardiovascular (HR = 1.32; 95% CI: 1.21, 1.44), and cancer (HR = 1.20; 95% CI: 1.12, 1.30) mortality (Table 2). The association between the metabolite score and all-cause mortality was consistent among women and men. Similarly, 30.5% of the association between the metabolite score and cardiovascular mortality could be attributed to incident CVD (Supplementary Fig. 2).

**Table 2 | Associations between the multi-metabolite profile score and all-cause, cardiovascular, and cancer mortality and longevity**

| | NHS/NHSII/HPFS[a] | | |
| --- | --- | --- | --- |
| | Cases/participants | Age-adjusted HR (95% CI)[b] | Multivariable-adjusted HR (95% CI)[b] |
| All-cause mortality | 4288/11,634 | 1.33 (1.28, 1.38) | 1.26 (1.22, 1.31) |
| Female | 3327/10,014 | 1.35 (1.29, 1.40) | 1.28 (1.22, 1.33) |
| Male | 961/1620 | 1.26 (1.15, 1.38) | 1.20 (1.09, 1.31) |
| Cardiovascular mortality | 867/11,634 | 1.42 (1.31, 1.54) | 1.32 (1.21, 1.44) |
| Cancer mortality | 1074/11,634 | 1.25 (1.17, 1.34) | 1.20 (1.12, 1.30) |
| | PREDIMED[c] | | |
| | Cases/participants | Age-adjusted HR (95% CI)[b] | Multivariable-adjusted HR (95% CI)[b] |
| All-cause mortality | 525/1878 | 1.48 (1.35, 1.62) | 1.39 (1.26, 1.53) |
| Cardiovascular mortality | 177/1878 | 1.57 (1.34, 1.82) | 1.45 (1.23, 1.71) |
| Cancer mortality | 149/1878 | 1.41 (1.18, 1.68) | 1.29 (1.07, 1.56) |
| | NHS/HPFS[d] | | |
| | Cases/participants | Age-adjusted OR (95% CI)[e] | Multivariable-adjusted OR (95% CI)[e] |
| Longevity | 3067/5297 | 0.71 (0.65, 0.76) | 0.77 (0.71, 0.83) |
| Female | 2527/4274 | 0.69 (0.64, 0.75) | 0.76 (0.69, 0.83) |
| Male | 540/1023 | 0.79 (0.67, 0.94) | 0.84 (0.70, 1.02) |
| | PREDIMED[f] | | |
| | Cases/participants | Age-adjusted OR (95% CI)[e] | Multivariable-adjusted OR (95% CI)[e] |
| Longevity | 458/865 | 0.71 (0.60, 0.84) | 0.80 (0.66, 0.95) |

[a]The age-adjusted model was stratified by study cohorts, original sub-studies, and the case/control status in the original sub-study and adjusted for age. Multivariable-adjusted model was further adjusted for fasting status, body mass index, race, multivitamin use, smoking status, physical activity, diabetes, hypertension, antihypertensive medication use, hypercholesterolemia, lipid-lowering medication use, total energy intake, alcohol intake, and Alternate Healthy Eating Index.
[b]Hazard ratio (HR) and 95% confidence interval (CI) of mortality risk per one standard deviation increment in the metabolite profile score, estimated using Cox models.
[c]The age-adjusted model was stratified by recruitment center and intervention group and adjusted for age. Multivariable model was further adjusted for sex, body mass index, smoking status, physical activity, diabetes, hypertension, antihypertensive medication use, hypercholesterolemia, lipid-lowering medication use, total energy intake, alcohol intake, and Mediterranean Diet Adherence Screener.
[d]The age-adjusted model was adjusted for study cohorts, original sub-studies, the case/control status in the original sub-study, and age. Multivariable model was further adjusted for fasting status, body mass index, race, multivitamin use, smoking status, physical activity, diabetes, hypertension, antihypertensive medication use, hypercholesterolemia, lipid-lowering medication use, total energy intake, alcohol intake, and Alternate Healthy Eating Index.
[e]Odds ratio (OR) and 95% confidence interval (CI) of longevity per one standard deviation increment in the metabolite profile score, estimated using logistic regression models.

## Metabolites and longevity

We further explored the associations between plasma metabolites and longevity, defined as reaching age 85 years. At the end of follow-up, 3067 participants achieved longevity. As expected, associations between metabolites and longevity were in the opposite direction of those for all-cause mortality (Pearson $r$ for β coefficients = −0.86) (Fig. 2e). We found 30 metabolites that were associated with longevity at FDR < 0.05 (Fig. 2e and Supplementary Data 6). Metabolites that were previously associated with lower mortality risk, such as L-serine and highly unsaturated TAGs and plasmalogens, were positively associated with longevity. In contrast, metabolites that were associated with increased risk of mortality, including N2,N2-dimethylguanosine, N1-acetylspermidine, N4-acetylcytidine, pseudouridine, and 4-acetamidobutanoic acid, were inversely associated with longevity. The associations for these 30 metabolites were similar among women and men and in the secondary analysis among men using the age threshold of 80 years to define longevity due to the shorter life expectancy in men (Supplementary Fig. 5). Associations of metabolite groups and modules for longevity were also in the opposite direction to those for all-cause mortality (Figs. 3a, c). The multi-metabolite profile score was associated with lower odds of achieving longevity (OR per 1-SD increment = 0.77; 95% CI: 0.71, 0.83), among both females and males (Table 2)

## External replication

We replicated the associations of metabolites with mortality and longevity in the PREDIMED. During a median follow-up of 14.5 years, 525 of 1878 participants died (177 cardiovascular deaths and 149 cancer deaths) and 458 reached longevity. We observed generally consistent associations for the 97 metabolites (Pearson $r$ for β coefficients = 0.79) that were associated with all-cause mortality in NHS/NHSII/HPFS and measured in PREDIMED (Supplementary Fig. 6). Using a criterion of raw $P < 0.05$ after multivariable adjustment, we replicated 20 out of 97 metabolite associations with all-cause mortality (Supplementary Data 7). These associations included inverse associations for highly unsaturated PCs and TAGs and positive associations for C16:0 ceramide (d18:1), pseudouridine, 4-acetamidobutanoic acid, N4-acetylcytidine, L-serine, and glycerolipids and phospholipids with less double bonds. We also observed positive associations of 4-acetamidobutanoic acid and C16:0 ceramide (d18:1) with cardiovascular mortality, and part of such associations was attributable to incident CVD (Supplementary Fig. 2). In addition, we replicated the positive associations of multi-metabolite profile score with all-cause (HR per 1-SD increment = 1.39; 95% CI: 1.26, 1.53), cardiovascular (HR = 1.45; 95% CI: 1.23, 1.71), and cancer (HR = 1.29; 95% CI: 1.07, 1.56) mortality, as well as the negative association with longevity (OR = 0.80; 95% CI: 0.66, 0.95) (Table 2).

## Discussion

Leveraging metabolomics data from 13,512 participants in four studies, we identified plasma metabolite profiles associated with all-cause mortality and longevity. Nucleosides including N2,N2-dimethylguanosine, pseudouridine, and N4-acetylcytidine, 4-acetamidobutanoic acid, N1-acetylspermidine, TAGs with ≤56 carbons and ≤3 double bonds, and DAGs, PCs, PEs, and plasmalogens with low saturation were strongly associated with increased risk of all-cause mortality and corresponding reduced likelihood of longevity; whereas L-serine, TAGs with >56 carbons or >3 double bonds, and highly unsaturated plasmalogens and PCs were associated with lower mortality risk and higher odds of achieving longevity. These associations were consistent in men and women and across different age strata. Moreover, we reported for the first time that 4-acetamidobutanoic acid and C16:0 Ceramide (d18:1) were associated with a higher risk of cardiovascular mortality, partially attributable to incident CVD. In addition, we developed a multi-metabolite profile score comprising 75 metabolites. A higher score was associated with increased all-cause, cardiovascular, and cancer mortality risk and decreased odds of longevity, which was robust across sex and in the external replication dataset.

Our results are consistent with previous findings for all-cause mortality and longevity from the Women's Health Initiative study[6] for three nucleosides related to post-transcriptional modifications of tRNA: N2,N2-dimethylguanosine, pseudouridine, and N4-acetylcytidine[13]. Elevated blood levels of N2,N2-dimethylguanosine and pseudouridine among patients with acute leukemia and breast cancer were first reported more than four decades ago[14]. This initial finding has been followed by a number of studies linking these two nucleosides and cancer, including colorectal cancer[15], ovarian cancer[16], and prostate cancer[17]. Recent studies extended these findings by showing that N2,N2-dimethylguanosine was also associated with high BMI[18]. Higher levels of N4-acetylcytidine were associated with higher insulinemic potential[19], and were found among older individuals with low-grade chronic inflammation[20] and patients with ovarian cancer[21] and other diseases[22]. Similar results in our sensitivity analysis where we excluded participants diagnosed with CVD and cancer within the first 4 years after blood collection gave us the reassurance that these associations were less likely due to reverse causation. Taken together, this evidence supports nucleoside metabolism as one potential biological pathway underpinning mortality and longevity.

We observed beneficial associations between two amino acids (L-serine and L-glutamine) and cardiovascular mortality and a harmful association for amino acid 4-acetamidobutanoic acid. The inverse association for L-serine in our analyses did not corroborate the previous study where higher serum serine was associated with an increased risk of all-cause mortality among hypertensive patients[23]. However, experimental studies reported an antihypertensive effect of L-serine, suggesting its favorable cardiovascular effects[24]. The associations for L-glutamine and 4-acetamidobutanoic acid were not obviously reported. In support of our findings, circulating glutamine has been inversely associated with multiple metabolic risk factors including insulin resistance and blood pressure[25]. Plasma 4-acetamidobutanoic acid was associated with incident heart failure[26] and a higher systemic immune-inflammation index (a CVD prognostic inflammatory marker)[27]. Our mediation analysis indicated that the association between 4-acetamidobutanoic acid and a higher risk of cardiovascular mortality could be largely attributed to incident CVD. Therefore, we speculated that 4-acetamidobutanoic acid is a key metabolite underlying CVD progression and thereby leads to an increased risk of cardiovascular death.

Little work has examined the associations of various lipid subclasses with mortality. One possible reason could be the differences in metabolite coverage across various metabolomic platforms. For example, nuclear magnetic resonance platforms tend to capture larger structures such as lipoproteins in detail, but they do not capture as much variety as mass spectrometry platforms[28]. By using high-throughput liquid chromatography-mass spectrometry platforms, we measured lipid metabolites from classes of TAGs, DAGs, CEs, PCs, PEs, plasmalogens, ceramides, carnitines, lysophospholipids, and sphingolipids. Some of the observed associations were consistent with previous reports. For example, higher C4-OH carnitine, a carnitine in pathways conferring increased risk of insulin resistance[29], was associated with increased risk of mortality in our study and the Women's Health Initiative study[6]. We additionally observed positive associations for L-acetylcarnitine and C14 carnitine. Higher levels of acylcarnitines are associated with fatty acid oxidation impairment and incident CVD[30,31]. We also found harmful associations for C16:0 Ceramide (d18:1) with mortality, especially cardiovascular mortality, which was partly attributable to CVD diagnosed after blood collection. Ceramides have been implicated in inflammation and proposed as new biomarkers of CVD[32,33]. Our findings extended their deleterious effects to cardiovascular death.

To our knowledge, our study reported for the first time that associations for PCs, plasmalogens, and TAGs with mortality depend on the number of carbons and double bonds in the acyl chains. Higher circulating PCs and plasmalogens with >6 double bonds and TAGs with >56 carbons or >3 double bonds were more likely to be associated with lower mortality risk and higher odds of achieving longevity, while those with fewer carbons and double bonds tended to have associations in the opposite direction. This pattern of associations was first reported for TAGs and type 2 diabetes[34]. Unsaturated lipids are more susceptible to peroxidation than saturated ones[35]. The favorable associations for highly unsaturated PCs, plasmalogens, and TAGs we observed suggested existence of enhanced antioxidant defenses. This is consistent with a previous study reporting that centenarians have lower plasma oxidative stress than their younger counterparts[36]. If further validated, these specific lipid species could be used to complement the standard clinical measurement of total TAGs.

In terms of origin and sources of the identified metabolites that were associated with mortality and longevity, our previous metabolomics analysis for plant-based diets observed that the three nucleosides (N2,N2-dimethylguanosine, pseudouridine, and N4-acetylcytidine) were positively associated with an unhealthy plant-based diet, specifically, the sugar-sweetened beverages component[37]. The positive association between N2,N2-dimethylguanosine and sugar-sweetened beverages has also been observed in another study among children[38]. Lipid metabolites, such as highly unsaturated TAGs, plasmalogens, and phospholipids, were positively associated with fish intake; whereas lipids including short-chain acylcarnitines and plasmalogens with less double bonds were positively associated with red meat consumption[39,40]. Besides dietary factors, genetic polymorphisms may also influence the levels of identified metabolites. For example, genetic variation on the *FADS* locus was strongly associated with long-chain polyunsaturated fatty acid-containing lipids[41]. Future studies integrating other omics technologies (e.g., genomics, transcriptomics) would provide more mechanistic insights into the pathways related to these metabolites.

Our study has several strengths, including the prospective examination of metabolites with mortality and longevity, the large sample size, the long-term follow-up, detailed covariable information, external replication in an independent dataset, and more importantly, the same metabolomics platform was used for both discovery and replication datasets. The opposite directions for the association of mortality and longevity reassured us of the validity of our metabolomics data. In addition, we conducted comprehensive analyses from individual metabolites to metabolite groups and to multi-metabolite profiles, and the results from these analyses corroborated each other. However, some limitations also deserve comment. The present analysis only included named metabolites from the mass spectrometry data. Future structural annotation of many other unknown peaks may

discover new biomarkers for mortality and longevity. Another limitation is that we cannot prove causal relationships due to the observational design of our study. Furthermore, we only collected blood samples at one time point for metabolomics measurement. Because the human metabolome is dynamic and constantly in flux, long-term repeated metabolomics data are needed to understand how changes in metabolite profiles or metabolite profiles at different time courses can predict mortality.

In summary, we identified a panel of nucleosides, amino acids, and lipid analytes associated with mortality and longevity, including two novel metabolites associated with the risk of cardiovascular mortality, namely, 4-acetamidobutanoic acid and C16:0 Ceramide (d18:1). We also observed a pattern that associations of several lipid subclasses with mortality were dependent on the number of double bonds, with a higher number being protective and lower number being harmful. In addition, we developed a metabolite profile score that was associated with an increased risk of mortality and decreased likelihood of longevity. Our results highlight some potentially important metabolites and biological pathways in aging and diseases that may open up new avenues to incorporate these metabolomic markers in clinical and research settings.

## Methods

### Study population

Our primary analyses were performed within three prospective US cohort studies: NHS, NHSII, and HPFS. The NHS started in 1976 and enrolled 121,700 female nurses aged 30–55 years[42]; the NHSII began in 1989 and recruited 116,429 female nurses aged 25–42 years[42]; and the HPFS was established in 1986 and enrolled 51,529 male health professionals aged 40–75 years[43]. Participants have been followed biennially and completed questionnaires obtaining lifestyle and health-related information. Blood samples were collected from 32,826 participants in the NHS during 1989–1990[44], from 29,611 participants in the NHSII during 1996–1999[44], and from 18,225 participants in the HPFS during 1993–1995[45] using a similar protocol. After collection, samples were shipped with an icepack via overnight courier to the laboratory, where they were processed and separated into plasma, buffy coat, and red blood cells and stored in liquid nitrogen freezers.

In the present study, we included participants from 13 prior nested case-control sub-studies on metabolomics (Supplementary Table 1). For the analysis of mortality, we excluded participants reporting a history of cardiovascular disease (CVD) and cancer at blood collection or lost to follow-up after blood draw. For the analysis of longevity, we excluded participants who were alive but did not achieve longevity (living to age 85 years) at the end of follow-up. In all, 11,634 participants were included in the prospective analysis for mortality and 5297 for longevity (Fig. 1). The study protocol for each cohort was approved by the institutional review boards (IRBs) of the Brigham and Women's Hospital, Harvard T.H. Chan School of Public Health, and participating registries. The present study was considered as non-human research by IRBs because it utilized de-identified samples that were originally collected for other studies.

The external replication for mortality was conducted in the PREDIMED study, a multicenter randomized controlled trial among individuals at high cardiovascular risk carried out in Spain from 2003 to 2009, which examined the effects of the traditional Mediterranean diet in the primary prevention of CVD, with type 2 diabetes as a secondary outcome[46,47]. For the present analyses, follow-up for death was available until 2020 through linkage with the National Death Index. We included participants from two nested case–cohort (CVD and diabetes outcomes) studies that included metabolomics profiling[31,48]. After excluding participants with >25% missing values in metabolites, 1878 were included in the replication analysis. The IRB of Hospital Clinic (Barcelona, Spain) approved the study protocol, and all participants provided written informed consent.

### Metabolomics measurement

The plasma metabolomics profiling in all four studies (NHS, NHSII, HPFS, and PREDIMED) was performed using high-throughput liquid chromatography-mass spectrometry (LC-MS) techniques at the Broad Institute of MIT and Harvard (Cambridge, MA) during 2015–2021[37,49,50]. Pooled plasma reference samples (prepared by combining small aliquots from the study samples), were analyzed every 20 participant samples to enable standardizing temporal drift in instrument response over time and between batches. In addition, quality control (QC) samples, to which the laboratory was blinded, were randomly distributed among the participants' samples and were also profiled.

Hydrophilic interaction liquid chromatography (HILIC) analyses of water-soluble metabolites in the positive ionization mode (HILIC-positive) were conducted using an LC–MS system comprised of a Shimadzu Nexera X2 U-HPLC (Shimadzu Corp.; Marlborough, MA) coupled to a Q Exactive mass spectrometer (Thermo Fisher Scientific; Waltham, MA). Metabolites were extracted from plasma (10 μL) using 90 μL of acetonitrile/methanol/formic acid (74.9:24.9:0.2 v/v/v) containing stable isotope-labeled internal standards (valine-d8, Sigma-Aldrich; St. Louis, MO; and phenylalanine-d8, Cambridge Isotope Laboratories; Andover, MA). The samples were centrifuged (10 min, $9000 \times g$, 4 °C), and the supernatants were injected directly onto a 150 × 2 mm, 3 μm Atlantis HILIC column (Waters; Milford, MA). The column was eluted isocratically at a flow rate of 250 μL/min with 5% mobile phase A (10 mM ammonium formate and 0.1% formic acid in water) for 0.5 min followed by a linear gradient to 40% mobile phase B (acetonitrile with 0.1% formic acid) over 10 min. MS analyses were carried out using electrospray ionization in the positive ion mode using full scan analysis over 70–800 $m/z$ at 70,000 resolution and 3 Hz data acquisition rate. Other MS settings were: sheath gas 40, sweep gas 2, spray voltage 3.5 kV, capillary temperature 350 °C, S-lens RF 40, heater temperature 300 °C, microscans 1, automatic gain control target 1e6, and maximum ion time 250 ms.

Plasma lipids were profiled using a Shimadzu Nexera X2 U-HPLC (Shimadzu Corp.; Marlborough, MA) (C8-positive). Lipids were extracted from plasma (10 μL) using 190 μL of isopropanol containing 1,2-didodecanoyl-sn-glycero-3-phosphocholine as an internal standard (Avanti Polar Lipids; Alabaster, AL). After centrifugation, supernatants were injected directly onto a 100 × 2.1 mm, 1.7 μm ACQUITY BEH C8 column (Waters; Milford, MA). The column was eluted isocratically with 80% mobile phase A (95:5:0.1 vol/vol/vol 10 mM ammonium acetate/methanol/formic acid) for 1 min followed by a linear gradient to 80% mobile-phase B (99.9:0.1 vol/vol methanol/formic acid) over 2 min, a linear gradient to 100% mobile phase B over 7 min, then 3 min at 100% mobile-phase B. MS analyses were carried out using electrospray ionization in the positive ion mode using full scan analysis over 200–1100 $m/z$ at 70,000 resolution and 3 Hz data acquisition rate. Other MS settings were: sheath gas 50, in source CID 5 eV, sweep gas 5, spray voltage 3 kV, capillary temperature 300 °C, S-lens RF 60, heater temperature 300 °C, microscans 1, automatic gain control target 1e6, and maximum ion time 100 ms.

Only named metabolites (a total of 396 measured in NHS/NHSII/HPFS) were considered in the present analysis. After excluding metabolites whose intraclass correlation coefficient across blinded quality control replicates (10% of study samples) <0.3 ($n = 8$) and metabolites with a missing rate ≥25% ($n = 145$), 243 metabolites were included in the current analysis. They were primarily lipids ($n = 171$, including 70 glycerolipids, 31 glycerophospholipids, 24 plasmalogens, 20 carnitines, 10 cholesterol esters [CEs], 8 lysophospholipids, 4 ceramides, and 4 sphingolipids), but also included amino acids related metabolites ($n = 31$), nucleosides and derivatives ($n = 12$), and other metabolites ($n = 29$). Most of the named metabolites (208 out of 243) were available for the replication analysis in PREDIMED. Metabolite data were log-transformed if highly skewed (absolute skewness > 2)[51], and all metabolites were then converted to z-scores within each sub-study. Missing

data for each metabolite were imputed using the random forest imputation approach as it has been previously recommended for metabolomics analysis[52].

## Ascertainment of mortality and longevity

Deaths in the cohorts were identified from state vital statistics records and the National Death Index or by reports from next of kin or the postal authorities. The follow-up for mortality in these cohorts is over 98% complete using these methods. The cause of death was determined by physician's review of medical records, autopsy reports, or death certificates. We used the International Classification of Diseases, Eighth Revision (ICD-8) in NHS and ICD-9 in HPFS, which were the ICD systems used at the time the cohorts began. Longevity was defined as living to the age of 85 years. This threshold has also been adopted by other studies[6]. In the PREDIMED study, four sources of information to identify endpoints were used: repeated contacts with participants, contacts with family physicians, a yearly review of medical records, and consultation with the National Death Index. All medical records that were related to endpoints were examined by medical doctors and ascertained by the end-point adjudication committee, whose members were unaware of the intervention-group assignments.

## Covariables

In NHS/NHSII/HPFS, we collected information on body weight, smoking status, physical activity, multivitamin use, race, diabetes, hypertension, hypercholesterolemia, antihypertensive medication use, and lipid-lowering medication use via self-reported questionnaires preceding blood collection. Body mass index (BMI) was calculated using height reported at cohort baseline and body weight reported before the blood draw. Age and fasting status were obtained through questionnaires completed at blood collection. Total calorie, alcohol intake, and the Alternate Healthy Eating Index (AHEI, a measure of overall diet quality ranging from 0–100, not including alcohol), were derived from the last semiquantitative food-frequency questionnaire (FFQ) before blood draw. The validity and reproducibility of FFQs have been reported elsewhere[53,54]. In PREDIMED, medical conditions, medication use, family history of diseases, and lifestyle factors were collected through a questionnaire during the first screening visit. At the baseline visit, trained personnel measured participants' body weight and height according to the study protocol.

## Statistical analysis

In NHS/NHSII/HPFS, the associations of individual metabolites (per 1-SD increment) with all-cause, cardiovascular, and cancer mortality were assessed by multivariable Cox regression models. The Cox regressions were stratified by cohort, original sub-study, and case/control status in the original sub-study and adjusted for age in months, BMI (continuous), race (white or non-white), fasting status (yes or no), multivitamin use (yes or no), smoking status (current, past, or never), physical activity (continuous), diabetes (yes or no), hypertension (yes or no), hypercholesterolemia (yes or no), antihypertensive medication use (yes or no), lipid-lowering medication use (yes or no), total energy intake (continuous), alcohol intake (continuous), and AHEI (continuous). The person-time for each participant was calculated from the blood collection date until the date of death or end of follow-up (June 2018 for the NHS/HPFS and June 2019 for the NHSII), whichever came first. We also conducted subgroup analyses for all-cause mortality by age at baseline (<55 or ≥55 years), sex, and case/control status and sensitivity analyses by limiting to the first 10- or 20-year follow-up and by excluding participants diagnosed with CVD or cancer within the first 4 years after blood collection. In addition, we conducted a mediation analysis to calculate the percentage of association with mortality that can be attributed to incident CVD.

We created knowledge-based metabolite groups with molecular or biological similarity and performed metabolite set enrichment analysis (MSEA)[55] to identify groups associated with mortality. MSEA ranks the metabolites by the multivariable-adjusted β coefficient of the association with mortality. This metric is used to identify enriched metabolite groups at the two extremes of the distribution of β estimates (positive/inverse associations)[16]. We derived metabolite modules using weighted gene co-expression network analysis (WGCNA), which constructs a scale-free network based on hierarchical clustering[16,56]. Each module was summarized by a score, calculated as the sum of measured metabolite values weighted by their corresponding loadings on the first principal component of all metabolites in that module. The module score was subsequently used in the Cox regression models to assess its association with mortality risk.

To account for the high correlations among metabolites and develop a multi-metabolite profile score for mortality, we used elastic net penalized Cox regression with a training-testing approach[57]. Participants were randomized to either the training set or the testing set in a 7 to 3 fashion. The elastic net model was first built in the training set within a 10-fold cross-validation framework and was then applied to the testing set to calculate the multi-metabolite profile score. The metabolite profile score was calculated as the weighted sum of the selected metabolites with weights equal to the elastic net regression coefficients. The score in the training set was obtained using a leave-one-out approach to avoid overfitting. We then examined the association between the metabolite score and mortality by multivariable Cox regression using combined data from the training set and testing set.

The associations between individual metabolites and longevity were evaluated by logistic regression with the same covariable adjustment as mortality. We also conducted a subgroup analysis by sex and sensitivity analysis for men by using 80 years instead of 85 years as the threshold due to the shorter life expectancy in men. The associations of knowledge-based metabolite groups, data-derived metabolite modules, and the multi-metabolite profile score with longevity were assessed as well. In the external replication dataset (PREDIMED Study) for all-cause and cardiovascular mortality, we conducted Cox regression analysis for available individual metabolites and the multi-metabolite profile score, stratified by recruitment center and intervention group and adjusted for age (continuous), sex (male or female), BMI (continuous), smoking status (current, past, or never), physical activity (continuous), baseline diabetes (yes or no), hypertension (yes or no), antihypertensive medication use (yes or no), hypercholesterolemia (yes or no), lipid-lowering medication use (yes or no), total energy intake (continuous), alcohol intake (continuous), and Mediterranean Diet Adherence Screener (MEDAS, a measure of overall diet quality in PREDIMED ranging from 0–14) (continuous). All statistical tests were two-sided. To account for multiple comparisons, we used the Benjamini–Hochberg procedure[58] and controlled the false discovery rate (FDR) < 5%. All analyses were performed in R version 4.1.0. The main R packages used were 'missRanger 2.1.5' for random forest imputation, 'survival 3.4-0' for Cox regression, 'fgsea 1.23.2' for MSEA, 'WGCNA 1.71' for metabolite module derivation, and 'glmnet 4.1-4' for elastic net regression.

## Reporting summary

Further information on research design is available in the Nature Portfolio Reporting Summary linked to this article.

# Data availability

Because of participant confidentiality and privacy concerns, data are available upon written request. According to standard controlled access procedure, applications to use NHS/NHSII/HPFS resources will be reviewed by our External Collaborators Committee for scientific aims, evaluation of the fit of the data for the proposed methodology, and verification that the proposed use meets the guidelines of the Ethics and Governance Framework and the consent that was provided by the participants. Investigators wishing to use NHS/NHSII/HPFS data are asked to submit a brief description of the proposed project

(contact: nhsaccess@channing.harvard.edu). Investigators can expect initial responses within 4 weeks of request submission. Details are available on https://www.nurseshealthstudy.org/researchers and https://sites.sph.harvard.edu/hpfs/for-collaborators/. Source data supporting all our findings (Figs. 2–3 and Supplementary Figs. 1–3 and 5–6) are provided with this publication as a Source Data file. Source data are provided with this paper.

## Code availability

The main code used to conduct this study is available on GitHub at https://github.com/fengleiwang/metabolomics_mortality_and_longevity.

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

## Acknowledgements

The authors thank the participants and staff of the NHS, NHSII, HPFS, and PREDIMED for their valuable contributions. The authors would like to acknowledge the contribution to this study from central cancer registries supported through the Centers for Disease Control and Prevention's National Program of Cancer Registries (NPCR) and/or the National Cancer Institute's Surveillance, Epidemiology, and End Results (SEER) Program. Central registries may also be supported by state agencies, universities, and cancer centers. Participating central cancer registries include the following: Alabama, Alaska, Arizona, Arkansas, California, Colorado, Connecticut, Delaware, Florida, Georgia, Hawaii, Idaho, Indiana, Iowa, Kentucky, Louisiana, Massachusetts, Maine, Maryland, Michigan, Mississippi, Montana, Nebraska, Nevada, New Hampshire, New Jersey, New Mexico, New York, North Carolina, North Dakota, Ohio, Oklahoma, Oregon, Pennsylvania, Puerto Rico, Rhode Island, Seattle SEER Registry, South Carolina, Tennessee, Texas, Utah, Virginia, West Virginia, Wyoming. This study is supported by the National Institutes of Health's grant 1R21AG070375 to M.G.-F. The NHS, NHSII, HPFS, and PREDIMED are supported by grants from the National Institutes of Health (UM1 CA186107, U01 CA176726, U01 CA167552, U01 HL145386, R01 CA49449, R01 HL034594, R01 HL088521, R01 CA67262, R01 HL35464, R01 HL60712, R01 CA50385, P01 CA87969, R01 AR049880, R01 DK112940, R01 DK119268, R01 DK120870, P30 DK046200, R01 HL118264, and R01 DK127601). F.W. is supported by the American Heart Association Postdoctoral Fellowship (Grant number: 897161). A.-J.T. is supported by the Canadian Institutes of Health Research Postdoctoral Fellowship Award (Grant number: 181882). M.G.-F. is supported by Novo Nordisk Foundation grant NNF18CC0034900. The funders had no role in the design, conduct, analysis, or reporting of this study. The funding sources did not participate in the design and conduct of the study; collection, management, analysis, and interpretation of the data; preparation, review, or approval of the manuscript; and decision to submit the manuscript for publication.

## Author contributions

F.W., F.B.H., and M.G.-F. designed the research. F.W. and A.-J.T. conducted analyses. A.H.E., C.C., and G.F.-D. participated in acquisition of data. L.L., K.H.L., and O.Z. provided statistical expertise. F.W. wrote the first draft of the paper. A.H.E., K.M.R., D.K.T., M.A.M.-G., J.S.-S., Q.S., M.J.S., F.B.H., and M.G.-F. obtained the funding. C.W., D.E.H., J.L., and F.G. contributed to the interpretation of the results and critical revision of the manuscript for important intellectual content. All authors approved the final version of the manuscript.

## Competing interests

The authors declare no competing interests.

## Additional information

[1]Department of Nutrition, Harvard T.H. Chan School of Public Health, Boston, MA, USA. [2]Department of Epidemiology, Harvard T.H. Chan School of Public Health, Boston, MA, USA. [3]Department of Biostatistics, Harvard T.H. Chan School of Public Health, Boston, MA, USA. [4]SciLifeLab, Division of Food Science and Nutrition, Department of Biology and Biological Engineering, Chalmers University of Technology, Gothenburg, Sweden. [5]Channing Division of Network Medicine, Brigham and Women's Hospital and Harvard Medical School, Boston, MA, USA. [6]Department of Preventive Medicine and Public Health, Navarra Health Research Institute (IDISNA), University of Navarra, Pamplona, Spain. [7]Division of Preventive Medicine, Department of Medicine, Brigham and Women's Hospital and Harvard Medical School, Boston, MA, USA. [8]Division of Women's Health, Department of Medicine, Brigham and Women's Hospital, and Harvard Medical School, Boston, MA, USA. [9]Rush Alzheimer's Disease Center, Rush University Medical Center, Chicago, IL, USA. [10]Consorcio CIBER, Fisiopatología de la Obesidad y Nutrición (CIBERObn), Instituto de Salud Carlos III (ISCIII), Madrid, Spain. [11]Universitat Rovira i Virgili, Departament de Bioquímica i Biotecnologia, Unitat de Nutrició Humana, Reus, Spain. [12]Institut d'Investigació Sanitària Pere Virgili (IISPV), Reus, Spain. [13]Metabolomics Platform, Broad Institute of MIT and Harvard, Cambridge, MA, USA. [14]Department of Public Health, Section of Epidemiology, University of Copenhagen, Copenhagen, Denmark. [15]Novo Nordisk Foundation Center for Basic Metabolic Research, University of Copenhagen, Copenhagen, Denmark. [16]These authors jointly supervised this work: Frank B. Hu, Marta Guasch-Ferré. ✉e-mail: fhu@hsph.harvard.edu; mguasch@hsph.harvard.edu

