## [Peer Review File · Nature Communications]

Plasma metabolomic profiles associated with mortality and longevity in a prospective analysis of 13,512 individualsREVIEWER COMMENTS

Reviewer #1 (Remarks to the Author):

This paper reported a prospective evaluation of plasma metabolite profiles for their associations with all-cause mortality, cardiovascular diseases (CVD), and cancer mortality, as well as longevity. Metabolomics data from 13 case-control/sub-studies nested in three well characterized cohort studies was included in the study. The external replication for mortality was conducted in the PREDIMED study where metabolic data was available from two nested case-cohorts for CVD and diabetes outcomes. The analysis was thorough and well thought out and the results were robust. The study revealed a few potentially important metabolites and biological pathways associated with aging and mortality. The paper was well written, and discussion and interpretation were appropriate. There is only a minor comment and two questions from this reviewer.

. While the results of sensitivity analyses presented in the supplement lend strong support to the robustness of study findings, some of the results, particularly the results of analysis from control participants, would be very informative and should be presented in detail in the main paper.

. What are the missing rates for the metabolites? Did the author conduct a complete data set analysis and compare the results from those generated with imputed data?

Reviewer #2 (Remarks to the Author):

The paper reports new associations between circulating metabolites and mortality/longevity, using metabolomics datasets from four large cohorts in the US and EU. Overall, the results provide new insights into how early metabolic changes may link to the aging process.

Strengths:

- All data were collected using the same untargeted metabolomics platform.
- Sample size: three important cohorts in the United States and an external validation cohort in Europe, with a total of 13,512 participants with untargeted metabolomics data.
- Median follow-up of 22.6 years in the US cohorts.
- Nucleotide and lipid metabolic pathways are linked with all-cause mortality and longevity.
- Focus is on older middle-aged (>55) people.

- 4414 total outcome events (deaths) in all four cohorts.
- Models were stratified by cohorts and sub-studies.

Weaknesses:

- FADS gene polymorphisms may explain the associations of PUFAs related lipids with mortality, which was not checked.
- Models need to be adjusted for fish oil (omega-3) intake.
- The metabolite-score has limited practical clinical utility because it requires 75 metabolites to compute. A simpler machine learning model using 3-5 significant metabolites will be more useful. Perhaps, one metabolite per WGCNA module can work.
- The discussion section lacks a mechanistic interpretation to cover the origin and sources (diet, organs, pathways, transport) of the mortality/longevity linked metabolites.
- The external validation cohort has very few outcome events (126) compared to 4288 in the main three cohorts, and a smaller follow-up of only 4.7 years.

Additional comments:

Abstract: Please mention the cohort names, geo-location, the number of total deaths, and the median follow-up duration.

Line 94: Add – in which year the metabolomics data were generated.

Line 95: How many unidentified metabolites were reported for these studies? Also, mention that the paper covers only the named metabolites.

Line 109: Was there any batch effect across the sub-studies? And how was it removed? Please add an unsupervised PCA plot of all cohorts to rule out that samples did or did not cluster by cohorts.

Line 211: Please add that these cohorts are from the US.

Line 230: Use "positive" and "negative" to indicate association directions, and "mortality" instead of "higher mortality"

Line 254: Mediation analyses for cancer and CVD incidences with overall mortality~metabolite are not provided.

Line 257: It will be interesting to see if 4-acetamidobutanoic acid at the time of CVD diagnosis can predict mortality.

Line 270: How many total modules were detected by WGCNA?

Line 274: What is a module score here? Sum or average?

Line 339: Rephrase the sentence to 'we first-time report that 4-acetamidobutanoic acid and C16:0 Ceramide (d18:1) were associated with.....' The paper reports 'new associations' not 'new metabolites'.

Line 381: Please discuss the differences in metabolomics assays and their compound lists as one of the key reasons why many studies have missed these chemical classes.

Line 402-03: Provide references.

Line 409: Justify why the score must be computed using 75 metabolites. For a routine clinical point of view, it will be more practical to have only 3-5 metabolites to compute this score.

Line 413-414: This sentence can be removed as results do not support it.

Line 423-425 – It is a core strength of this analysis that data for all four studies were generated using the same platform. Please mention that.

Line 423-425 - Remove these two sentences. "The metabolomics platform that generated our data targeted cationic metabolites, including amino acids and lipid species. Metabolomics profiling of other molecular classes such as metabolites involved in carbohydrate metabolism may reveal additional metabolites associated with mortality and longevity." It is unnecessary since it remains true for any metabolomics assay given the fact that a single assay cannot cover the chemical diversity of the metabolome. This also contradicts the use of phrase "metabolome-wide association analysis".

Instead, please add that the mass spectrometry data were collected in an untargeted mode and there were many LC/MS peaks that were unidentified. Structural annotation of these peaks may discover new biomarkers for mortality and longevity.

REVIEWER COMMENTS

Reviewer #1 (Remarks to the Author):

This paper reported a prospective evaluation of plasma metabolite profiles for their associations with all-cause mortality, cardiovascular diseases (CVD), and cancer mortality, as well as longevity. Metabolomics data from 13 case-control/sub-studies nested in three well characterized cohort studies was included in the study. The external replication for mortality was conducted in the PREDIMED study where metabolic data was available from two nested case-cohorts for CVD and diabetes outcomes. The analysis was thorough and well thought out and the results were robust. The study revealed a few potentially important metabolites and biological pathways associated with aging and mortality. The paper was well written, and discussion and interpretation were appropriate. There is only a minor comment and two questions from this reviewer.

- We thank reviewer for the positive evaluation of our manuscript.

. While the results of sensitivity analyses presented in the supplement lend strong support to the robustness of study findings, some of the results, particularly the results of analysis from control participants, would be very informative and should be presented in detail in the main paper.

- As suggested, we have moved the results by case/control status to the main figure (Figure 2b) and emphasized it in the main text (Page 6, lines 77-79).

. What are the missing rates for the metabolites? Did the author conduct a complete data set analysis and compare the results from those generated with imputed data?

- We excluded metabolites with missing rates >25%. The missing rates for the included metabolites ranged from 0-24.7%. We also conducted a complete data set analysis. The results were almost the same as those using random forest to impute missingness (Pearson $r=0.9986$ for two sets β beta coefficients). We have included this result in Supplementary Figure 1f.

Reviewer #2 (Remarks to the Author):

The paper reports new associations between circulating metabolites and mortality/longevity, using metabolomics datasets from four large cohorts in the US and EU. Overall, the results provide new insights into how early metabolic changes may link to the aging process.

- We appreciate the reviewer's comments that helped to improve our manuscript.

Strengths:

- All data were collected using the same untargeted metabolomics platform.
- Sample size: three important cohorts in the United States and an external validation cohort in Europe, with a total of 13,512 participants with untargeted metabolomics data.
- Median follow-up of 22.6 years in the US cohorts.
- Nucleotide and lipid metabolic pathways are linked with all-cause mortality and longevity.

- Focus is on older middle-aged (>55) people.
- 4414 total outcome events (deaths) in all four cohorts.
- Models were stratified by cohorts and sub-studies.
- We thank reviewer for these positive comments.

Weaknesses:

- FADS gene polymorphisms may explain the associations of PUFAs related lipids with mortality, which was not checked.
- Thank you for the comment. We agree that genetic polymorphisms may help explain the associations of metabolites with mortality, such as *FADS* genes for PUFA-related lipids. However, conducting genetic analyses in the present study would be beyond the scope of this paper. We have incorporated genetic polymorphisms into the Discussion section. We will consider including genetic data in future studies.

On page 15, lines 280-285, we state:

“Besides dietary factors, genetic polymorphisms may also influence the levels of identified metabolites. For example, genetic variation on the FADS locus was strongly associated with long-chain polyunsaturated fatty acid-containing lipids.⁴¹ Future studies integrating other omics technologies (e.g., genomics, transcriptomics) would provide more mechanistic insights into the pathways related to these metabolites.”

- Models need to be adjusted for fish oil (omega-3) intake.
- We adjusted for AHEI (Alternate Healthy Eating Index, a measure of overall diet quality ranging from 0-100) to account for the overall diet quality. The AHEI already includes ratio of polyunsaturated to saturated fatty acid ratio as one component.
- As suggested, we also ran a model further adjusting for omega-3 intake. The results were almost the same as those from the model without adjusting for omega-3 intake (Pearson $r = 0.9999$ for two sets of beta coefficients).
- The metabolite-score has limited practical clinical utility because it requires 75 metabolites to compute. A simpler machine learning model using 3-5 significant metabolites will be more useful. Perhaps, one metabolite per WGCNA module can work.
- Thank you for the comment. We agree that a metabolite score comprising fewer metabolites would be more useful in clinical settings. However, mortality is a complex outcome and cannot be fully captured by a few metabolites. We tried to select different numbers of metabolites with larger weights and calculated new metabolite scores. We then examined the associations between these metabolite scores and mortality/longevity. The results indicated that a minimum of 35-45 metabolites were required for the metabolite score to achieve comparable associations as the score including all 75 metabolites (selected by the elastic net regression). When using LC-MS to measure metabolites, the cost for measuring 35-45 metabolites is similar to that for 75 metabolites. Therefore, we kept the metabolite score including 75 metabolites in our paper because of the potential bias of focusing only in 3-5 metabolites.

Association between metabolite scores comprising different numbers of metabolites and mortality/longevity

	NHS/NHSII/HPFS		PREDIMED	
	All-cause mortality	Longevity	All-cause mortality	Longevity
n=5 (top 5)	1.15 (1.11, 1.19)	0.84 (0.77, 0.90)	1.27 (1.16, 1.40)	0.81 (0.68, 0.96)
n=21 (weight>0.05)	1.24 (1.19, 1.29)	0.79 (0.73, 0.85)	1.36 (1.23, 1.49)	0.82 (0.68, 0.98)
n=35 (weight>0.03)	1.27 (1.22, 1.32)	0.77 (0.71, 0.83)	1.37 (1.24, 1.51)	0.81 (0.67, 0.96)
n=45 (weight>0.02)	1.28 (1.23, 1.33)	0.76 (0.70, 0.82)	1.39 (1.26, 1.53)	0.79 (0.66, 0.94)
n=58 (weight>0.01)	1.28 (1.23, 1.33)	0.76 (0.70, 0.82)	1.39 (1.26, 1.53)	0.79 (0.66, 0.95)
n=75 (all selected metabolites)	1.26 (1.22, 1.31)	0.77 (0.71, 0.83)	1.39 (1.26, 1.53)	0.80 (0.66, 0.95)

The weights for 75 selected metabolites ranged from -0.11 to 0.11.

- The discussion section lacks a mechanistic interpretation to cover the origin and sources (diet, organs, pathways, transport) of the mortality/longevity linked metabolites.
- We have added one paragraph in the Discussion section to cover the origin and sources of the identified metabolites (Page 15, lines 271-285).

The new paragraph reads:

“In terms of origin and sources of the identified metabolites that were associated with mortality and longevity, our previous metabolomics analysis for plant-based diets observed that the three nucleosides (N2,N2-dimethylguanosine, pseudouridine, and N4-acetylcytidine) were positively associated with an unhealthy plant-based diet, specifically, the sugar-sweetened beverages component.³⁷ The positive association between N2,N2-dimethylguanosine and sugar-sweetened beverages has also been observed in another study among children.³⁸ Lipid metabolites, such as highly unsaturated TAGs, plasmalogens, and phospholipids, were positively associated with fish intake; whereas lipids including short-chain acylcarnitines and plasmalogens with less double bonds were positively associated with red meat consumption.^{39,40} Besides dietary factors, genetic polymorphisms may also influence the levels of identified metabolites. For example, genetic variation on the FADS locus was strongly associated with long-chain polyunsaturated fatty acid-containing lipids.⁴¹ Future studies integrating other omics technologies (e.g., genomics, transcriptomics) would provide more mechanistic insights into the pathways related to these metabolites.”

- The external validation cohort has very few outcome events (126) compared to 4288 in the main three cohorts, and a smaller follow-up of only 4.7 years.
- Thank you for the comment. Recently, the external validation cohort (PREDIMED) updated follow-up information in March 2023. Now, we have been able to replicate our results in this external validation cohort with a longer follow-up (median 14.5 years). The total death records increased by more than 4-fold from 126 to 525 based on linkage with the National Death Index up to 2020. We are now also able to replicate the longevity results. We believe that the addition of this new data will help

to address the valid concern of the Reviewer. We have provided the updated results in Table 2 and Supplementary Data 7.

Additional comments:

Abstract: Please mention the cohort names, geo-location, the number of total deaths, and the median follow-up duration.

- Due to the word limit, we did not mention the study names, but have included country, the number of total deaths, and the median follow-up duration in the abstract. It reads as follows (Page 3, lines 3-7):

“Here we examine the associations of 243 plasma metabolites with mortality and longevity (attaining age 85 years) in 11,634 US (median follow-up of 22.6 years, with 4288 deaths) and 1878 Spanish participants (median follow-up of 14.5 years, with 525 deaths).”

Line 94: Add – in which year the metabolomics data were generated.

- Done. The metabolomics data was generated during 2015-2021.

Line 95: How many unidentified metabolites were reported for these studies? Also, mention that the paper covers only the named metabolites.

- The number of unknown metabolites (peaks) reported for different sub-studies ranged from ~2000 to ~5000. As suggested, we have revised our description and mentioned that this paper only covered the named metabolites. It reads as follows (Page 19, lines 358-360):

“Only named metabolites (a total of 396 measured in NHS/NHSII/HPFS) were considered in the present analysis.”

Line 109: Was there any batch effect across the sub-studies? And how was it removed? Please add an unsupervised PCA plot of all cohorts to rule out that samples did or did not cluster by cohorts.

- As mentioned, the plasma metabolomic profiling was conducted over a period of several years. To address the batch effect, pooled plasma reference samples (prepared by combining small aliquots from the study samples), were analyzed every 20 participant samples to enable standardizing temporal drift in instrument response over time and between batches. In addition, quality control (QC) samples, to which the laboratory was blinded, were randomly distributed among the participants' samples and were also profiled. After obtaining the metabolite data from the laboratory, we converted the metabolite data to z-scores within each sub-study to further minimize the influence of batch effect.
- As suggested, we made unsupervised PCA plots and did not observe clustering by cohorts and sub-studies.

Line 211: Please add that these cohorts are from the US.

- We have added it in the Methods section. It reads as follows:

“Our primary analyses were performed within three prospective US cohort studies: NHS, NHSII, and HPFS.”

Line 230: Use "positive" and "negative" to indicate association directions, and "mortality" instead of "higher mortality"

- As suggested, we have changed to use “positive” and “negative” to report the association directions for mortality.

Line 254: Mediation analyses for cancer and CVD incidences with overall mortality~metabolite are not provided.

- The mediation results for CVD incidences with all-cause mortality and metabolite were provided in Supplementary Data 2. The percentage explained by incident CVD was <10% for almost all the metabolites. We also have included the mediation results for cancer incidences in Supplementary Data 3. The percentage explained by incident cancer was < 10% for all metabolites. Mediation results are also described in lines 107-108 (page 8) of the Results section.

Line 257: It will be interesting to see if 4-acetamidobutanoic acid at the time of CVD diagnosis can predict mortality.

- Thank you for the comment. We agree that it would be interesting to assess the association between metabolites like 4-acetamidobutanoic acid at the time of CVD diagnosis and future mortality risk. Unfortunately, we only have metabolite data at one time point, before disease diagnosis. We have mentioned this in the limitation section. It reads as follows (page 16, lines 299-303):

“Furthermore, we only collected blood samples at one time point for metabolomics measurement. Because the human metabolome is dynamic and constantly in flux, long-term repeated metabolomics data are needed to understand how changes in metabolite profiles or metabolite profiles at different time courses can predict mortality.”

Line 270: How many total modules were detected by WGCNA?

- WGCNA detected 5 modules. It reads as follows in the Results section (page 8, lines 119-120):

“WGCNA identified five metabolite modules.”

Line 274: What is a module score here? Sum or average?

- It is the sum of metabolite values weighted by their corresponding loadings on the first principal component of all metabolites in that module. We have clarified it in the Methods section (page 22, lines 431-433):

“Each module was summarized by a score, calculated as the sum of measured metabolite values weighted by their corresponding loadings on the first principal component of all metabolites in that module.”

Line 339: Rephrase the sentence to 'we first-time report that 4-acetamidobutanoic acid and C16:0 Ceramide (d18:1) were associated with.....' The paper reports 'new associations' not 'new metabolites'.

- As suggested, we have rephrased the sentence. Now it reads as follows (page 11, lines 196-198):

“Moreover, we reported for the first-time that 4-acetamidobutanoic acid and C16:0 Ceramide (d18:1) were associated with a higher risk of cardiovascular mortality.”

Line 381: Please discuss the differences in metabolomics assays and their compound lists as one of the key reasons why many studies have missed these chemical classes.

- Thank you for the comment. We have added that the difference in metabolomic platforms could be one reason why little work has examined various lipid subclass. It reads as follows (page 13, lines 238-242):

“One possible reason could be the differences in metabolite coverage across various metabolomic platforms. For example, nuclear magnetic resonance platforms tend to capture larger structures such as lipoproteins in detail, but they do not capture as much variety as mass spectrometry platforms.”

Line 402-03: Provide references.

- We have the provided reference indicating that unsaturated lipids are more susceptible to peroxidation than saturated ones.

Line 409: Justify why the score must be computed using 75 metabolites. For a routine clinical point of view, it will be more practical to have only 3-5 metabolites to compute this score.

- The 75 metabolites were selected by agnostic approaches using the elastic net regression.

- As mentioned above, we agree that a metabolite score comprising fewer metabolites would be more useful in clinical settings. However, mortality is a complex outcome and cannot be fully captured by a few metabolites. We tried selecting different numbers of metabolites with larger weights and calculated new metabolite scores. We then examined the associations between these metabolite scores and mortality/longevity. The results indicated that a minimum of 35-45 metabolites were required for the metabolite score to achieve comparable associations as the score including all 75 metabolites (selected by the elastic net regression). When using LC-MS to measure metabolites, the cost for measuring 35-45 metabolites will be similar to that for 75 metabolites. Therefore, we kept the metabolite score including 75 metabolites in our paper. Moreover, the main aim of this manuscript was to identify a multi-metabolite profile that captures several pathways to predict mortality.

Line 413-414: This sentence can be removed as results do not support it.

- We have removed this sentence.

Line 423-425 – It is a core strength of this analysis that data for all four studies were generated using the same platform. Please mention that.

- Thank you for the comment. We have included this as one strength. It reads as follows (page 15, lines 287-291):

“Our study has several strengths, including the prospective examination of metabolites with mortality and longevity, the large sample size, the long-term follow-up, detailed covariable information, external replication in an independent dataset, and more importantly, the same metabolomics platform was used for both discovery and replication datasets.”

Line 423-425 - Remove these two sentences. “The metabolomics platform that generated our data targeted cationic metabolites, including amino acids and lipid species. Metabolomics profiling of other molecular classes such as metabolites involved in carbohydrate metabolism may reveal additional metabolites associated with mortality and longevity.” It is unnecessary since it remains true for any metabolomics assay given the fact that a single assay cannot cover the chemical diversity of the metabolome. This also contradicts the use of phrase “metabolome-wide association analysis”. Instead, please add that the mass spectrometry data were collected in an untargeted mode and there were many LC/MS peaks that were unidentified. Structural annotation of these peaks may discover new biomarkers for mortality and longevity.

- Thank you for the comments. We have removed the two sentences and added that the current analysis only included named metabolites and that future annotation of unknown peaks may discover new biomarkers for mortality and longevity. It reads as follows (page 16, lines 295-298):

“The present analysis only included named metabolites from the mass spectrometry data. Future structural annotation of many other unknown peaks may discover new biomarkers for mortality and longevity.”

REVIEWERS' COMMENTS

Reviewer #1 (Remarks to the Author):

The revised manuscript has adequately addressed all concerns of the previous review.

Reviewer #2 (Remarks to the Author):

Authors have made extensive and very satisfactory edits in the manuscript in response to my comments. I have no more concerns.

Reviewer #1 (Remarks to the Author):

The revised manuscript has adequately addressed all concerns of the previous review.

- We thank reviewer for the comments.

Reviewer #2 (Remarks to the Author):

Authors have made extensive and very satisfactory edits in the manuscript in response to my comments. I have no more concerns.

- We thank reviewer for the comment.